# Finite Versus Infinite Neural Networks: an Empirical Study

**Jaehoon Lee**    **Samuel S. Schoenholz**[*]    **Jeffrey Pennington**[*]    **Ben Adlam**[*†]

**Lechao Xiao**[*]    **Roman Novak**[*]    **Jascha Sohl-Dickstein**

Google Brain
{jaehlee, schsam, jpennin, adlam, xlc, romann, jaschasd}@google.com

## Abstract

We perform a careful, thorough, and large scale empirical study of the correspondence between wide neural networks and kernel methods. By doing so, we resolve a variety of open questions related to the study of infinitely wide neural networks. Our experimental results include: kernel methods outperform fully-connected finite-width networks, but underperform convolutional finite width networks; neural network Gaussian process (NNGP) kernels frequently outperform neural tangent (NT) kernels; centered and ensembled finite networks have reduced posterior variance and behave more similarly to infinite networks; weight decay and the use of a large learning rate break the correspondence between finite and infinite networks; the NTK parameterization outperforms the standard parameterization for finite width networks; diagonal regularization of kernels acts similarly to early stopping; floating point precision limits kernel performance beyond a critical dataset size; regularized ZCA whitening improves accuracy; finite network performance depends non-monotonically on width in ways not captured by double descent phenomena; equivariance of CNNs is only beneficial for narrow networks far from the kernel regime. Our experiments additionally motivate an improved layer-wise scaling for weight decay which improves generalization in finite-width networks. Finally, we develop improved best practices for using NNGP and NT kernels for prediction, including a novel ensembling technique. Using these best practices we achieve state-of-the-art results on CIFAR-10 classification for kernels corresponding to each architecture class we consider.

## 1  Introduction

A broad class of both Bayesian [1–17] and gradient descent trained [13–16, 18–29] neural networks converge to Gaussian Processes (GPs) or closely-related kernel methods as their intermediate layers are made infinitely wide. The predictions of these infinite width networks are described by the Neural Network Gaussian Process (NNGP) [4, 5] kernel for Bayesian networks, and by the Neural Tangent Kernel (NTK) [18] and weight space linearization [24, 25] for gradient descent trained networks.

This correspondence has been key to recent breakthroughs in our understanding of neural networks [30–40]. It has also enabled practical advances in kernel methods [8, 9, 15, 16, 26, 41–43], Bayesian deep learning [44–46], active learning [47], and semi-supervised learning [17]. The NNGP, NTK, and related large width limits [10, 30, 48–66] are unique in giving an exact theoretical description of large scale neural networks. Because of this, we believe they will continue to play a transformative role in deep learning theory.

---

[*]SSS, JP, BA, LX, and RN contributed equally.

[†]Work done as a member of the Google AI Residency program (https://g.co/airesidency).

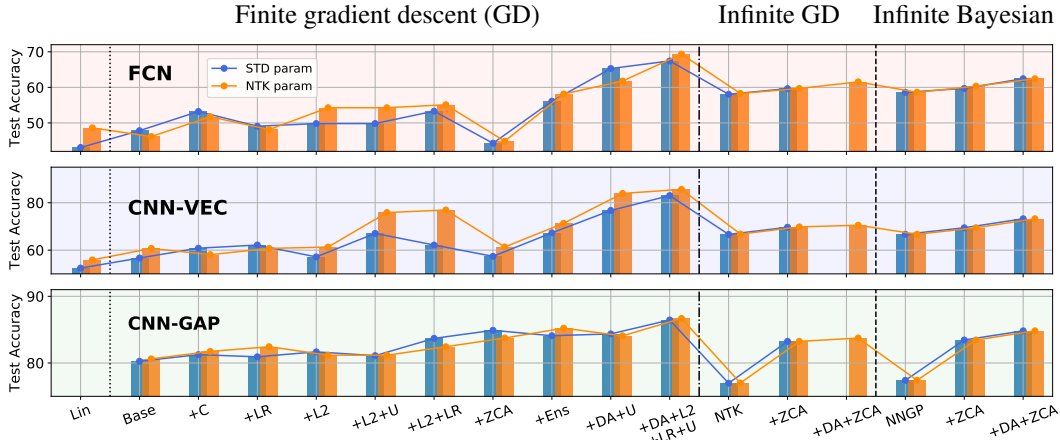

Figure 1: **CIFAR-10 test accuracy for finite and infinite networks and their variations**. Starting from the finite width `base` network of given architecture class described in §2, performance changes from **centering** (+C), **large learning rate** (+LR), allowing **underfitting** by early stopping (+U), input preprocessing with **ZCA regularization** (+ZCA), multiple initialization **ensembling** (+Ens), and some combinations are shown, for **Standard** and **NTK** parameterizations. The performance of the **linearized** (`lin`) base network is also shown. See Table S1 for precise values for each of these experiments, as well as for additional experimental conditions not shown here.

Infinite networks are a newly active field, and foundational empirical questions remain unanswered. In this work, we perform an extensive and in-depth empirical study of finite and infinite width neural networks. In so doing, we provide quantitative answers to questions about the factors of variation that drive performance in finite networks and kernel methods, uncover surprising new behaviors, and develop best practices that improve the performance of both finite and infinite width networks. We believe our results will both ground and motivate future work in wide networks.

To learn more about the infinite-width limit, we refer to the general references cited above. As starting points, we believe Novak et al. [9] provides the clearest introduction to the NNGP, and Lee et al. [24] provides the clearest introduction to the NTK.

## 2 Experiment design

To systematically develop a phenomenology of infinite and finite neural networks, we first establish base cases for each architecture where infinite-width kernel methods, linearized weight-space networks, and nonlinear gradient descent based training can be directly compared. In the finite-width settings, the base case uses mini-batch gradient descent at a constant small learning rate [24] with MSE loss (implementation details in §H). In the kernel-learning setting we compute the NNGP and NTK for the entire dataset and do exact inference as described in [67, page 16]. Once this one-to-one comparison has been established, we augment the base setting with a wide range of interventions. We discuss each of these interventions in detail below. Some interventions will approximately preserve the correspondence (for example, data augmentation), while others explicitly break the correspondence in a way that has been hypothesized in the literature to affect performance (for example, large learning rates [39]). We additionally explore linearizing the base model around its initialization, in which case its training dynamics become exactly described by a constant kernel. This differs from the kernel setting described above due to finite width effects.

We use MSE loss to allow for easier comparison to kernel methods, whose predictions can be evaluated in closed form for MSE. See Table S2 and Figure S3 for a comparison of MSE to softmax-cross-entropy loss. Softmax-cross-entropy provides a consistent small benefit over MSE, and will be interesting to consider in future work.

Architectures we work with are built from either Fully-Connected (`FCN`) or Convolutional (`CNN`) layers. In all cases we use ReLU nonlinearities with critical initialization with small bias variance ($\sigma_w^2 = 2.0, \sigma_b^2 = 0.01$). Except if otherwise stated, we consider `FCN`s with 3-layers of width 2048 and `CNN`s with 8-layers of 512 channels per layer. For convolutional networks we must collapse the spatial dimensions of image-shaped data before the final readout layer. To do this we either:

flatten the image into a one-dimensional vector (VEC) or apply global average pooling to the spatial dimensions (GAP). Finally, we compare two ways of parameterizing the weights and biases of the network: the standard parameterization (STD), which is used in work on finite-width networks, and the NTK parameterization (NTK) which has been used in most infinite-width studies to date (see [27] for the standard parameterization at infinite width).

Except where noted, for all kernel experiments we optimize over diagonal kernel regularization independently for each experiment. For finite width networks, except where noted we use a small learning rate corresponding to the base case. See §C.1 for details.

The experiments described in this paper are often very compute intensive. For example, to compute the NTK or NNGP for the entirety of CIFAR-10 for CNN-GAP architectures one must explicitly evaluate the entries in a $6 \times 10^7$-by-$6 \times 10^7$ kernel matrix due to pixel-pixel covariances. Typically this takes around 1200 GPU hours with double precision, and so we implement our experiments via massively distributed compute infrastructure based on beam [68]. All experiments use the Neural Tangents library [15], built on top of JAX [69].

To be as systematic as possible while also tractable given this large computational requirement, we evaluated every intervention for every architecture and focused on a single dataset, CIFAR-10 [70]. However, to ensure robustness of our results across dataset, we evaluate several key claims on CIFAR-100 and Fashion-MNIST [71].

## 3 Observed empirical phenomena

### 3.1 NNGP/NTK can outperform finite networks

A common assumption in the study of infinite networks is that they underperform the corresponding finite network in the large data regime. We carefully examine this assumption, by comparing kernel methods against the base case of a finite width architecture trained with small learning rate and no regularization (§2), and then individually examining the effects of common training practices which break (large LR, L2 regularization) or improve (ensembling) the infinite width correspondence to kernel methods. The results of these experiments are summarized in Figure 1 and Table S1.

First focusing on base finite networks, we observe that infinite FCN and CNN-VEC outperform their respective finite networks. On the other hand, infinite CNN-GAP networks perform worse than their finite-width counterparts in the base case, consistent with observations in Arora et al. [26]. We emphasize that architecture plays a key role in relative performance, in line with an observation made in Geiger et al. [61] in the study of lazy training. For example, infinite-FCNs outperform finite-width networks even when combined with various tricks such as high learning rate, L2, and underfitting. Here the performance becomes similar only after ensembling (§3.3).

One interesting observation is that ZCA regularization preprocessing (§3.10) can provide significant improvements to the CNN-GAP kernel, closing the gap to within 1-2%.

### 3.2 NNGP typically outperforms NTK

Recent evaluations of infinite width networks have put significant emphasis on the NTK, without explicit comparison against the respective NNGP models [26, 29, 41, 42]. Combined with the view of NNGPs as "weakly-trained" [24, 26] (i.e. having only the last layer learned), one might expect NTK to be a more effective model class than NNGP. On the contrary, we usually observe that NNGP inference achieves better performance. This can be seen in Table S1 where SOTA performance among fixed kernels is attained with the NNGP across all architectures. In Figure 2 we show that this trend persists across CIFAR-10, CIFAR-100, and Fashion-MNIST (see Figure S5 for similar trends on UCI regression tasks). In addition to producing stronger models, NNGP kernels require about half the memory and compute as the corresponding NTK, and some of the most performant kernels do not have an associated NTK at all [43]. Together these results suggest that when approaching a new problem where the goal is to maximize performance, practitioners should start with the NNGP.

We emphasize that both tuning of the diagonal regularizer (Figure 5) and sufficient numerical precision (§3.7, Figure S1) were crucial to achieving an accurate comparison of these kernels.

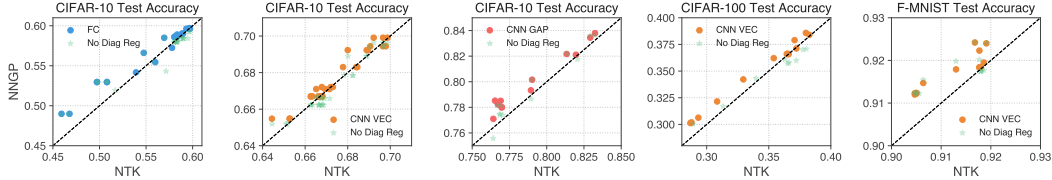

Figure 2: **NNGP often outperforms NTK in image classification tasks when diagonal regularization is carefully tuned.** The performance of the NNGP and NT kernels are plotted against each other for a variety of data pre-processing configurations (§3.10), while regularization (Figure 5) is independently tuned for each.

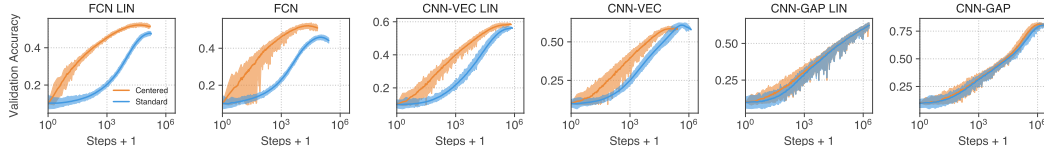

Figure 3: **Centering can accelerate training and improve performance.** Validation accuracy throughout training for several finite width architectures. See Figure S6 for training accuracy.

## 3.3 Centering and ensembling finite networks both lead to kernel-like performance

For overparameterized neural networks, some randomness from the initial parameters persists throughout training and the resulting learned functions are themselves random. This excess variance in the network's predictions generically increases the total test error through the variance term of the bias-variance decomposition. For infinite-width kernel systems this variance is eliminated by using the mean predictor. For finite-width models, the variance can be large, and test performance can be significantly improved by *ensembling* a collection of models [61, 62]. In Figure 4, we examine the effect of ensembling. For FCN, ensembling closes the gap with kernel methods, suggesting that finite width FCNs underperform FCN kernels primarily due to variance. For CNN models, ensembling also improves test performance, and ensembled CNN-GAP models significantly outperform the best kernel methods. The observation that ensembles of finite width CNNs can outperform infinite width networks while ensembles of finite FCNs cannot (see Figure 4) is consistent with earlier findings in [62].

Prediction variance can also be reduced by *centering* the model, i.e. subtracting the model's initial predictions: $f_{\text{centered}}(t) = f(\theta(t)) - f(\theta(0))$. A similar variance reduction technique has been studied in [25, 72–74]. In Figure 3, we observe that centering significantly speeds up training and improves generalization for FCN and CNN-VEC models, but has little-to-no effect on CNN-GAP architectures. We observe that the scale posterior variance of CNN-GAP, in the infinite-width kernel, is small relative to the prior variance given more data, consistent with centering and ensembles having small effect.

## 3.4 Large LRs and L2 regularization drive differences between finite networks and kernels

In practice, L2 regularization (a.k.a. weight decay) or larger learning rates can break the correspondence between kernel methods and finite width neural network training even at large widths.

Lee et al. [24] derives a critical learning rate $\eta_{\text{critical}}$ such that wide network training dynamics are equivalent to linearized training for $\eta < \eta_{\text{critical}}$. Lewkowycz et al. [39] argues that even at large width a learning rate $\eta \in (\eta_{\text{critical}}, c \cdot \eta_{\text{critical}})$ for a constant $c > 1$ forces the network to move away from its initial high curvature minimum and converge to a lower curvature minimum, while Li et al. [75]

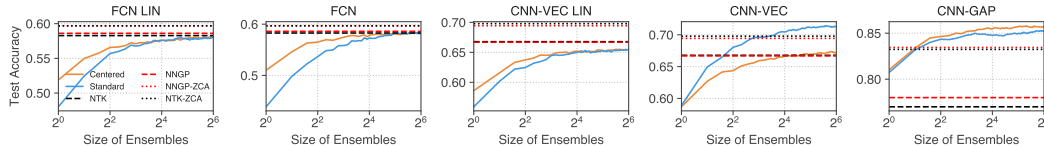

Figure 4: **Ensembling base networks enables them to match the performance of kernel methods, and exceed kernel performance for nonlinear CNNs.** See Figure S7 for test MSE.

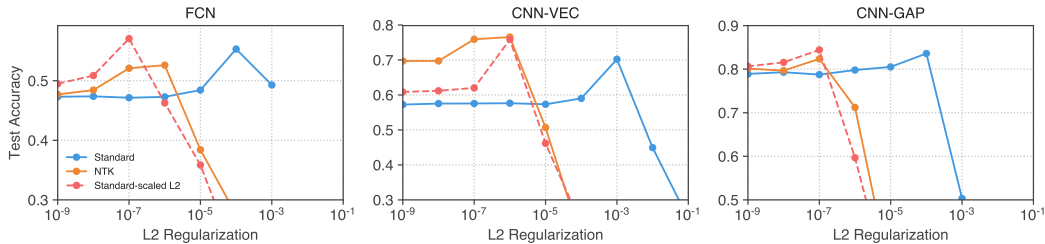

Figure 5: **Layerwise scaling motivated by NTK makes L2 regularization more helpful in standard parameterization networks.** See §3.5 for introduction of the improved regularizer, Figure S9 for further analysis on L2 regularization to initial weights, and Figure S8 for effects on varying widths.

argues that large initial learning rates enable networks to learn 'hard-to-generalize' patterns.

In Figure 1 (and Table S1), we observe that the effectiveness of a large learning rate (LR) is highly sensitive to both architecture and paramerization: LR improves performance of `FCN` and `CNN-GAP` by about 1% for STD parameterization and about 2% for NTK parameterization. In stark contrast, it has little effect on `CNN-VEC` with NTK parameterization and surprisingly, a huge performance boost on `CNN-VEC` with STD parameterization ($+5\%$).

L2 regularization (Equation S1) regularizes the squared distance between the parameters and the origin and encourages the network to converge to minima with smaller Euclidean norms. Such minima are different from those obtained by NT kernel-ridge regression (i.e. adding a diagonal regularization term to the NT kernel) [32], which essentially penalizes the deviation of the network's parameters from initialization [76]. See Figure S8 for a comparison.

L2 regularization consistently improves ($+1\text{-}2\%$) performance for all architectures and parameterizations. Even with a well-tuned L2 regularization, finite width `CNN-VEC` and `FCN` still underperform NNGP/NTK. Combining L2 with early stopping produces a dramatic additional $10\% - 15\%$ performance boost for finite width `CNN-VEC`, outperforming NNGP/NTK. Finally, we note that L2+LR together provide a superlinear performance gain for all cases except `FCN` and `CNN-GAP` with NTK-parameterization. Understanding the nonlinear interactions between L2, LR, and early stopping on finite width networks is an important research question (e.g. see [39, 40] for LR/L2 effect on the training dynamics).

## 3.5 Improving L2 regularization for networks using the standard parameterization

We find that L2 regularization provides dramatically more benefit (by up to 6%) to finite width networks with the NTK parameterization than to those that use the standard parameterization (see Table S1). There is a bijective mapping between weights in networks with the two parameterizations, which preserves the function computed by both networks: $W_{\text{STD}}^l = W_{\text{NTK}}^l / \sqrt{n^l}$, where $W^l$ is the $l$th layer weight matrix, and $n^l$ is the width of the preceding activation vector. Motivated by the improved performance of the L2 regularizer in the NTK parameterization, we use this mapping to construct a regularizer for standard parameterization networks that produces the same penalty as vanilla L2 regularization would produce on the equivalent NTK-parameterized network. This modified regularizer is $R_{\text{Layerwise}}^{\text{STD}} = \frac{\lambda}{2} \sum_l n^l \|W_{\text{STD}}^l\|^2$. This can be thought of as a layer-wise regularization constant $\lambda^l = \lambda n^l$. The improved performance of this regularizer is illustrated in Figure 5.

## 3.6 Performance can be non-monotonic in width beyond double descent

Deep learning practitioners have repeatedly found that increasing the number of parameters in their models leads to improved performance [9, 77–82]. While this behavior is consistent with a Bayesian perspective on generalization [83–85], it seems at odds with classic generalization theory which primarily considers worst-case overfitting [86–92]. This has led to a great deal of work on the interplay of overparameterization and generalization [93–102]. Of particular interest has been the phenomenon of double descent, in which performance increases overall with parameter account, but drops dramatically when the neural network is roughly critically parameterized [103–105].

Empirically, we find that in most cases (`FCN` and `CNN-GAP` in both parameterizations, `CNN-VEC`

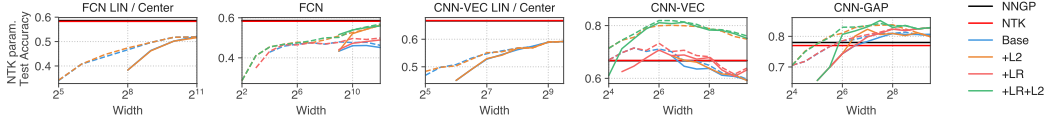

Figure 6: **Finite width networks generally perform better with increasing width, but CNN-VEC shows surprising non-monotonic behavior. L2**: non-zero weight decay allowed during training **LR**: large learning rate allowed. Dashed lines are allowing underfitting (**U**). See Figure S10 for plots for the standard parameterization, and §3.11 for discussion of CNN-VEC results.

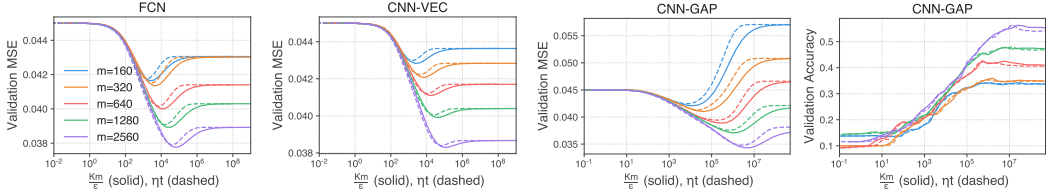

Figure 7: **Diagonal kernel regularization acts similarly to early stopping.** Solid lines corresponds to NTK inference with varying diagonal regularization $\varepsilon$. Dashed lines correspond to predictions after gradient descent evolution to time $\tau = \eta t$ (with $\eta = m/\mathrm{tr}(\mathcal{K})$). Line color indicates varying training set size $m$. Performing early stopping at time $t$ corresponds closely to regularizing with coefficient $\varepsilon = Km/\eta t$, where $K = 10$ denotes number of output classes.

with standard parameterization) increasing width leads to monotonic improvements in performance. However, we also find a more complex dependence on width in specific relatively simple settings. For example, in Figure 6 for CNN-VEC with NTK parameterization the performance depends non-monotonically on the width, and the optimal width has an intermediate value.[3] This nonmonotonicity is distinct from double-descent-like behavior, as all widths correspond to overparameterized models.

### 3.7 Diagonal regularization of kernels behaves like early stopping

When performing kernel inference, it is common to add a diagonal regularizer to the training kernel matrix, $\mathcal{K}_{\mathrm{reg}} = \mathcal{K} + \varepsilon \frac{\mathrm{tr}(\mathcal{K})}{m} I$. For linear regression, Ali et al. [108] proved that the inverse of a kernel regularizer is related to early stopping time under gradient flow. With kernels, gradient flow dynamics correspond directly to training of a wide neural network [18, 24].

We experimentally explore the relationship between early stopping, kernel regularization, and generalization in Figure 7. We observe a close relationship between regularization and early stopping, and find that in most cases the best validation performance occurs with early stopping and non-zero $\varepsilon$. While Ali et al. [108] do not consider a $\frac{\mathrm{tr}(\mathcal{K})}{m}$ scaling on the kernel regularizer, we found it useful since experiments become invariant under scale of $\mathcal{K}$.

### 3.8 Floating point precision determines critical dataset size for failure of kernel methods

We observe empirically that kernels become sensitive to `float32` vs. `float64` numerical precision at a critical dataset size. For instance, GAP models suffer `float32` numerical precision errors at a dataset size of $\sim 10^4$. This phenomena can be understood with a simple random noise model (see §D for details). The key insight is that kernels with fast eigenvalue decay suffer from floating point noise. Empirically, the tail eigenvalue of the NNGP/NTK follows a power law (see Figure 8) and measuring their decay trend provides good indication of critical dataset size

$$ m^* \gtrsim \left(C/(\sqrt{2}\sigma_n)\right)^{\frac{2}{2\alpha-1}} \quad \text{if } \alpha > \tfrac{1}{2} \qquad (\infty \quad \text{otherwise}) \,, \tag{1} $$

where $\sigma_n$ is the typical noise scale, e.g. `float32` epsilon, and the kernel eigenvalue decay is modeled as $\lambda_i \sim C\,i^{-\alpha}$ as $i$ increases. Beyond this critical dataset size, the smallest eigenvalues in the kernel become dominated by floating point noise.

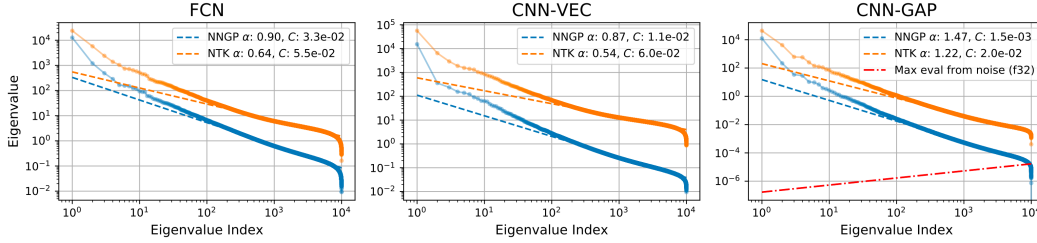

Figure 8: **Tail eigenvalues of infinite network kernels show power-law decay.** The red dashed line shows the predicted scale of noise in the eigenvalues due to floating point precision, for kernel matrices of increasing width. Eigenvalues for CNN-GAP architectures decay fast, and may be overwhelmed by `float32` quantization noise for dataset sizes of $O(10^4)$. For `float64`, quantization noise is not predicted to become significant until a dataset size of $O(10^{10})$ (Figure S1).

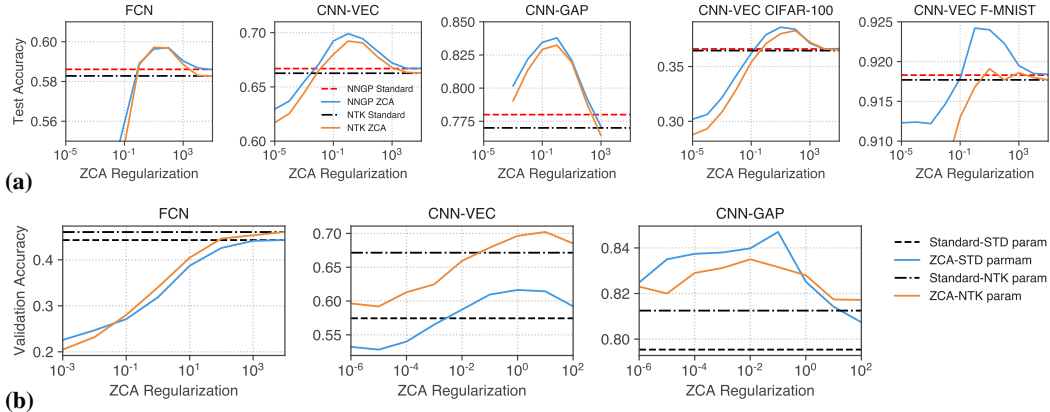

Figure 9: **Regularized ZCA whitening improves image classification performance for both finite and infinite width networks.** All plots show performance as a function of ZCA regularizaiton strength. (**a**) ZCA whitening of inputs to kernel methods on CIFAR-10, Fashion-MNIST, and CIFAR-100. (**b**) ZCA whitening of inputs to finite width networks (training curves in Figure S11).

### 3.9 Linearized `CNN-GAP` models perform poorly due to poor conditioning

We observe that the linearized `CNN-GAP` converges *extremely* slowly on the training set (Figure S6), leading to poor validation performance (Figure 3). Even after training for more than 10M steps with varying L2 regularization strengths and LRs, the best training accuracy was below 90%, and test accuracy ~70% – worse than both the corresponding infinite and nonlinear finite width networks.

This is caused by poor conditioning of pooling networks. Xiao et al. [33] (Table 1) show that the conditioning at initialization of a `CNN-GAP` network is worse than that of `FCN` or `CNN-VEC` networks by a factor of the number of pixels (1024 for CIFAR-10). This poor conditioning of the kernel eigenspectrum can be seen in Figure 8. For linearized networks, in addition to slowing training by a factor of 1024, this leads to numerical instability when using `float32`.

### 3.10 Regularized ZCA whitening improves accuracy

ZCA whitening [109] (see Figure S2 for an illustration) is a data preprocessing technique that was once common [110, 111], but has fallen out of favor. However it was recently shown to dramatically improve accuracy in some kernel methods by Shankar et al. [43], in combination with a small regularization parameter in the denominator (see §F). We investigate the utility of ZCA whitening as a preprocessing step for both finite and infinite width neural networks. We observe that while pure ZCA whitening is detrimental for both kernels and finite networks (consistent with predictions in [112]), with tuning of the regularization parameter it provides performance benefits for both kernel methods and finite network training (Figure 9).

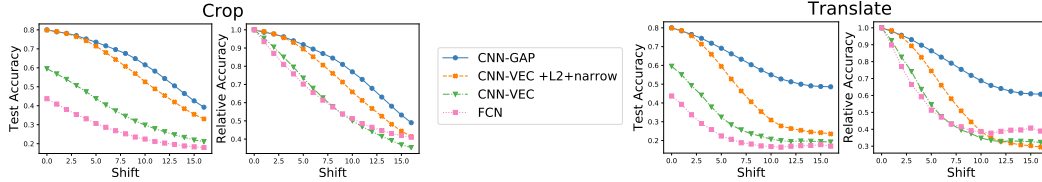

Figure 10: **Equivariance is only leveraged in a `CNN` model outside of the kernel regime.** If a `CNN` model is able to utilize equivariance effectively, we expect it to be more robust to crops and translations than an `FCN`. Surprisingly, performance of a wide `CNN-VEC` degrades with the magnitude of the input perturbation as fast as that of an `FCN`, indicating that equivariance is not exploited. In contrast, performance of a narrow model with weight decay (`CNN-VEC+L2+narrow`) falls off much slower. Translation-invariant `CNN-GAP` remains, as expected, the most robust. Details in §3.11, §C.1.

## 3.11 Equivariance is only beneficial for narrow networks far from the kernel regime

Due to weight sharing between spatial locations, outputs of a convolutional layer are translation-*equivariant* (up to edge effects), i.e. if an input image is translated, the activations are translated in the same spatial direction. However, the vast majority of contemporary CNNs utilize weight sharing in conjunction with pooling layers, making the network outputs approximately translation-*invariant* (`CNN-GAP`). The impact of equivariance alone (`CNN-VEC`) on generalization is not well understood – it is a property of internal representations only, and does not translate into meaningful statements about the classifier outputs. Moreover, in the infinite-width limit it is guaranteed to have no impact on the outputs [9, 13]. In the finite regime it has been reported both to provide substantial benefits by Novak et al. [9], Lecun [113] and no significant benefits by Bartunov et al. [114].

We conjecture that equivariance can only be leveraged far from the kernel regime. Indeed, as observed in Figure 1 and discussed in §3.4, multiple kernel correspondence-breaking tricks are required for a meaningful boost in performance over NNGP or NTK (which are mathematically guaranteed to not benefit from equivariance), and the boost is largest at a moderate width (Figure 6). Otherwise, even large ensembles of equivariant models (see `CNN-VEC LIN` in Figure 4) perform comparably to their infinite width, equivariance-agnostic counterparts. Accordingly, prior work that managed to extract benefits from equivariant models [9, 113] tuned networks far outside the kernel regime (extremely small size and +LR+L2+U respectively). We further confirm this phenomenon in a controlled setting in Figure 10.

## 3.12 Ensembling kernel predictors enables practical data augmentation with NNGP/NTK

Finite width neural network often are trained with data augmentation (DA) to improve performance. We observe that the `FCN` and `CNN-VEC` architectures (both finite and infinite networks) benefit from DA, and that DA can cause `CNN-VEC` to become competitive with `CNN-GAP` (Table S1). While `CNN-VEC` possess translation equivariance but not invariance (§3.11), we believe it can effectively leverage equivariance to learn invariance from data.

For kernels, expanding a dataset with augmentation is computationally challenging, since kernel computation is quadratic in dataset size, and inference is cubic. Li et al. [41], Shankar et al. [43] incorporated flip augmentation by doubling the training set size. Extending this strategy to more augmentations such as crop or mixup [115], or to broader augmentations strategies like AutoAugment [116] and RandAugment [117], becomes rapidly infeasible.

Here we introduce a straightforward method for ensembling kernel predictors to enable more extensive data augmentation. More sophisticated approximation approaches such as the Nyström method [118] might yield even better performance. The strategy involves constructing a set of augmented batches, performing kernel inference for each of them, and then performing ensembling of the resulting predictions. This is equivalent to replacing the kernel with a block diagonal approximation, where each block corresponds to one of the batches, and the union of all augmented batches is the full augmented dataset. See §E for more details. This method achieves SOTA for a kernel method corresponding to the infinite width limit of each architecture class we studied (Figure 11 and Table 1).

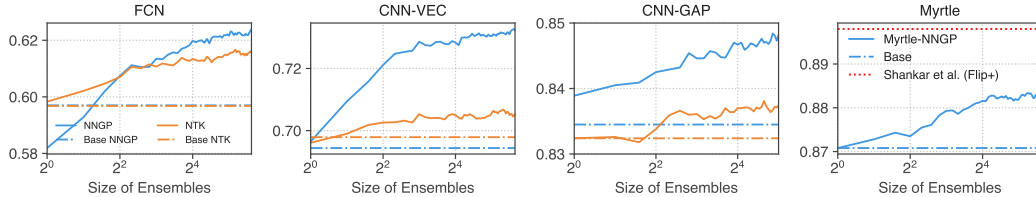

Figure 11: **Ensembling kernel predictors makes predictions from large augmented datasets computationally tractable.** We used standard crop by 4 and flip data augmentation (DA) common for training neural networks for CIFAR-10. We observed that DA ensembling improves accuracy and is much more effective for NNGP compared to NTK. In the last panel, we applied data augmentation by ensemble to the Myrtle architecture studied in Shankar et al. [43]. We observe improvements over our base setting, but do not reach the reported best performance. We believe techniques such as leave-one-out tilt and ZCA augmentation also used in [43] contribute to this difference.

Table 1: **CIFAR-10 test accuracy for kernels of the corresponding architecture type**

| Architecture | Method | NTK | NNGP |
|---|---|---|---|
| **FC** | Novak et al. [9] | - | 59.9 |
| | ZCA Reg (this work) | 59.7 | 59.7 |
| | DA Ensemble (this work) | **61.5** | **62.4** |
| **CNN-VEC** | Novak et al. [9] | **-** | 67.1 |
| | Li et al. [41] | 66.6 | 66.8 |
| | ZCA Reg (this work) | 69.8 | 69.4 |
| | Flip Augmentation, Li et al. [41] | 69.9 | 70.5 |
| | DA Ensemble (this work) | **70.5** | **73.2** |
| **CNN-GAP** | Arora et al. [26], Li et al. [41] | 77.6 | 78.5 |
| | ZCA Reg (this work) | 83.2 | 83.5 |
| | Flip Augmentation, Li et al. [41] | 79.7 | 80.0 |
| | DA Ensemble (this work) | **83.7 (32 ens)** | **84.8 (32 ens)** |
| **Myrtle** [4] | Myrtle ZCA and Flip Augmentation, Shankar et al. [43] | - | **89.8** |

## 4   Discussion

We performed an in-depth investigation of the phenomenology of finite and infinite width neural networks through a series of controlled interventions. We quantified phenomena having to do with generalization, architecture dependence, deviations between infinite and finite networks, numerical stability, data augmentation, data preprocessing, ensembling, network topology, and failure modes of linearization. We further developed best practices that improve performance for both finite and infinite networks. We believe our experiments provide firm empirical ground for future studies.

The careful study of other architectural components such as self-attention, normalization, and residual connections would be an interesting extension to this work, especially in light of results such as Goldblum et al. [119] which empirically observes that the large width behavior of Residual Networks does not conform to the infinite-width limit. Another interesting future direction would be incorporating systematic finite-width corrections, such as those in Antognini [63], Dyer and Gur-Ari [64], Huang and Yau [65], Yaida [66].

## Broader Impact

Developing theoretical understanding of neural networks is crucial both for understanding their biases, and predicting when and how they will fail. Understanding biases in models is of critical importance if we hope to prevent them from perpetuating and exaggerating existing racial, gender, and other social biases [120–123]. Understanding model failure has a direct impact on human safety, as neural networks increasingly do things like drive cars and control the electrical grid [124–126].

We believe that wide neural networks are currently the most promising direction for the development

of neural network theory. We further believe that the experiments we present in this paper will provide empirical underpinnings that allow better theory to be developed. We thus believe that this paper will in a small way aid the engineering of safer and more just machine learning models.

## Acknowledgments and Disclosure of Funding

We thank Yasaman Bahri and Ethan Dyer for discussions and feedback on the project. We are also grateful to Atish Agarwala and Gamaleldin Elsayed for providing valuable feedback on a draft.

We acknowledge the Python community [127] for developing the core set of tools that enabled this work, including NumPy [128], SciPy [129], Matplotlib [130], Pandas [131], Jupyter [132], JAX [133], Neural Tangents [15], Apache Beam [68], Tensorflow datasets [134] and Google Colaboratory [135].

No external funding or competing interest related to this work.

## Footnotes

[3]Similar behavior was observed in [106] for CNN-VEC and in [107] for finite width Bayesian networks.

[4]The normalized Gaussian Myrtle kernel used in Shankar et al. [43] does not have a corresponding finite-width neural network, and was additionally tuned on the test set for the case of CIFAR-10.

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
