[Supplementary Material]

# Supplementary Material

## A  Glossary

We use the following abbreviations in this work:

- **L2**: L2 reguarization a.k.a. weight decay;
- **LR**: using large learning rate;
- **U**: allowing underfitting;
- **DA**: using data augmentation;
- **C**: centering the network so that the logits are always zero at initialization;
- **Ens**: neural network ensembling logits over multiple initialization;
- **ZCA**: zero-phase component analysis regularization preprocessing;
- **FCN**: fully-connected neural network.;
- **CNN-VEC**: convolutional neural network with a vectorized readout layer;
- **CNN-GAP**: convolutional neural network with a global average pooling readout layer;
- **NNGP**: neural network Gaussian process;
- **NTK**: neural tangent kernel.

## B  Main table

Table S1: **CIFAR-10 classification accuracy for nonlinear and linearized finite neural networks, as well as for NTK and NNGP kernel methods**. Starting from `Base` network of given architecture class described in §2, performance change of **centering** (+C), **large learning rate** (+LR), allowing **underfitting** by early stopping (+U), input preprocessing with **ZCA regularization** (+ZCA), multiple initialization **ensembling** (+Ens), and some combinations are shown, for Standard and NTK parameterization. See also Figure 1.

|  | Param | Base | +C | +LR | +L2 | +L2 +U | +L2 +LR | +L2 +LR +U | +ZCA | Best w/o DA | +Ens | +Ens +C | +DA +U | +DA +L2 +LR +U |
|---|---|---|---|---|---|---|---|---|---|---|---|---|---|---|
| FCN | STD | 47.82 | 53.22 | 49.07 | 49.82 | 49.82 | 55.32 | 55.32 | 44.29 | 55.90 | 58.11 | 58.25 | 65.29 | 67.43 |
| | NTK | 46.16 | 51.74 | 48.14 | 54.27 | 54.27 | 55.11 | 55.44 | 44.86 | 55.44 | 58.14 | 58.31 | 61.87 | 69.35 |
| CNN-VEC | STD | 56.68 | 60.82 | 62.16 | 57.15 | 67.07 | 62.16 | 68.99 | 57.39 | 68.99 | 67.30 | 65.65 | 76.73 | 83.01 |
| | NTK | 60.73 | 58.09 | 60.73 | 61.30 | 75.85 | 76.93 | 77.47 | 61.35 | 77.47 | 71.32 | 67.23 | 83.92 | 85.63 |
| CNN-GAP | STD | 80.26 | 81.25 | 80.93 | 81.67 | 81.10 | 83.69 | 83.01 | 84.90 | 84.22 | 84.15 | 84.62 | 84.36 | 86.45 |
| | NTK | 80.61 | 81.73 | 82.44 | 81.17 | 81.17 | 82.44 | 82.43 | 83.75 | 83.92 | 85.22 | 85.75 | 84.07 | 86.68 |

|  | Param | Lin Base | +C | +L2 | +L2 +U | +Ens | +Ens +C | NTK | +ZCA | +DA +ZCA | NNGP | +ZCA | +DA +ZCA |
|---|---|---|---|---|---|---|---|---|---|---|---|---|---|
| FCN | STD | 43.09 | 51.48 | 44.16 | 50.77 | 57.85 | 57.99 | 58.05 | 59.65 | - | 58.61 | 59.70 | 62.40 |
| | NTK | 48.61 | 52.12 | 51.77 | 51.77 | 58.04 | 58.16 | 58.28 | 59.68 | 61.54 | | | |
| CNN-VEC | STD | 52.43 | 60.61 | 58.41 | 58.41 | 64.58 | 64.67 | 66.64 | 69.65 | - | 66.69 | 69.44 | 73.23 |
| | NTK | 55.88 | 58.94 | 58.52 | 58.50 | 65.45 | 65.54 | 66.78 | 69.79 | 70.52 | | | |
| CNN-GAP | STD | >70.00* (Train accuracy 86.22 after 14M steps) | | | | | | 76.97 | 83.24 | - | 78.0 | 83.45 | 84.82 |
| | NTK | >68.59* (Train accuracy 79.90 after 14M steps) | | | | | | 77.00 | 83.24 | 83.74 | | | |

## C  Experimental details

For all experiments, we use Neural Tangents (NT) library [15] built on top of JAX [133]. First we describe experimental settings that is mostly common and then describe specific details and hyperparameters for each experiments.

**Finite width neural networks** We train finite width networks with Mean Squared Error (MSE) loss

$$\mathcal{L} = \frac{1}{2|\mathcal{D}|K} \sum_{(x_i, y_i) \in \mathcal{D}} \|f(x_i) - y_i\|^2 \,,$$

where $K$ is the number of classes and $\| \cdot \|$ is the $L^2$ norm in $\mathbb{R}^K$. For the experiments with +L2, we add L2 regularization to the loss

$$R_{\text{L2}} = \frac{\lambda}{2} \sum_l \|W^l\|^2\,,\tag{S1}$$

and tune $\lambda$ using grid-search optimizing for the validation accuracy.

We optimize the loss using mini-batch SGD with constant learning rate. We use batch-size of 100 for `FCN` and 40 for both `CNN-VEC` and `CNN-GAP` (see §H for further details on this choice). Learning rate is parameterized with learning rate factor $c$ with respect to the critical learning rate

$$\eta = c\,\eta_{\text{critical}}\,.\tag{S2}$$

In practice, we compute empirical NTK $\hat{\Theta}(x, x') = \sum_j \partial_j f(x) \partial_j f(x')$ on 16 random points in the training set to estimate $\eta_{\text{critical}}$ [24] by maximum eigenvalue of $\hat{\Theta}(x, x)$. This is readily available in NT library [15] using `nt.monte_carlo_kernel_fn` and `nt.predict.max_learning_rate`. Base case considered without large learning rate indicates $c \leq 1$, and large learning rate (+LR) runs are allowing $c > 1$. Note that for linearized networks $\eta_{\text{critical}}$ is strict upper-bound for the learning rates and no $c > 1$ is allowed [24, 36, 39].

Training steps are chosen to be large enough, such that learning rate factor $c \leq 1$ can reach above $99\%$ accuracy on $5k$ random subset of training data for 5 logarithmic spaced measurements. For different learning rates, physical time $t = \eta \times$ (# of steps) roughly determines learning dynamics and small learning rate trials need larger number of steps. Achieving termination criteria was possible for all of the trials except for linearized `CNN-GAP` and data augmented training of `FCN`, `CNN-VEC`. In these cases, we report best achieved performance without fitting the training set.

**NNGP / NTK** For inference, except for data augmentation ensembles for which default zero regularization was chosen, we grid search over diagonal regularization in the range `numpy.logspace(-7, 2, 14)` and 0. Diagonal regularization is parameterized as

$$\mathcal{K}_{\text{reg}} = \mathcal{K} + \varepsilon \frac{\text{tr}(\mathcal{K})}{m} I$$

where $\mathcal{K}$ is either NNGP or NTK for the training set. We work with this parameterization since $\varepsilon$ is invariant to scale of $\mathcal{K}$.

**Dataset** For all our experiments (unless specified) we use train/valid/test split of 45k/5k/10k for CIFAR-10/100 and 50k/10k/10k for Fashion-MNIST. For all our experiments, inputs are standardized with per channel mean and standard deviation. ZCA regularized whitening is applied as described in §F. Output is encoded as mean subtracted one-hot-encoding for the MSE loss, e.g. for a label in class $c$, $-0.1 \cdot \mathbf{1} + \mathbf{e_c}$. For the softmax-cross-entropy loss in §G, we use standard one-hot-encoded output.

For data augmentation, we use widely-used augmentation for CIFAR-10; horizontal flips with $50\%$ probability and random crops by 4-pixels with zero-padding.

**Details of architecture choice:** We only consider ReLU activation (with the exception of Myrtle-kernel which use scaled Gaussian activation [43]) and choose critical initialization weight variance of $\sigma_w^2 = 2$ with small bias variance $\sigma_b^2 = 0.01$. For convolution layers, we exclusively consider $3 \times 3$ filters with stride 1 and `SAME` (zero) padding so that image size does not change under convolution operation.

### C.1 Hyperparameter configurations for all experiments

We used grid-search for tuning hyperparameters and use accuracy on validation set for deciding on hyperparameter configuration or measurement steps (for underfitting / early stopping). All reported numbers unless specified is test set performance.

**Figure 1, Table S1**: We grid-search over L2 regularization strength $\lambda \in \{0\} \cup \{10^{-k}|k$ from -9 to -3$\}$ and learning rate factor $c \in \{2^k|k$ from -2 to 5$\}$. For linearized networks same search space is used except that $c > 1$ configuration is infeasible and training diverges. For non-linear, centered runs $c \in \{2^k|k$ from 0 to 4$\}$ is used. Network ensembles uses base configuration with $\lambda = 0$, $c = 1$ with 64 different initialization seed. Kernel ensemble is over 50 predictors for `FCN` and `CNN-VEC` and 32 predictors for `CNN-GAP`. Finite networks trained with data-augmentation has different learning rate factor range of $c \in \{1, 4, 8\}$.

**Figure 2**: Each datapoint corresponds to either standard preprocessed or ZCA regularization preprocessed (as described in §3.10) with regularization strength was varied in $\{10^{-k}|k \in [-6, -5, ..., 4, 5]\}$ for FCN and CNN-VEC, $\{10^{-k}|k \in [-3, -2, ..., 2, 3]\}$ for CNN-GAP.

**Figure 3, Figure 4, Figure S6, Figure S7**: Learning rate factors are $c = 1$ for non-linear networks and $c = 0.5$ for linearized networks. While we show NTK parameterized runs, we also observe similar trends for STD parameterized networks. Shaded regions show range of minimum and maximum performance across 64 different seeds. Solid line indicates the mean performance.

**Figure 5** While FCN is the base configuration, CNN-VEC is a narrow network with 64 channels per layer since moderate width benefits from L2 more for the NTK parameterization Figure S10. For CNN-GAP 128 channel networks is used. All networks with different L2 strategy are trained with +LR ($c > 1$).

**Figure 6, Figure S8, Figure S10**: $\lambda \in \{0, 10^{-9}, 10^{-7}, 10^{-5}, 10^{-3}\}$ and $c \in \{2^k|k \text{ from } -2 \text{ to } 5\}$.

**Figure 7**: We use 640 subset of validation set for evaluation. CNN-GAP is a variation of the base model with 3 convolution layers with $\sigma_b^2 = 0.1$ while FCN and CNN-VEC is the base model. Training evolution is computed using analytic time-evolution described in Lee et al. [24] and implemented in NT library via nt.predict.gradient_descent_mse with 0 diagonal regularization.

**Figure 9**: Kernel experiments details are same as in Figure 2. Finite networks are base configuration with $c = 1$ and $\lambda = 0$.

**Figure 10**: Evaluated networks uses NTK parameterization with $c = 1$. CNN-VEC+L2+narrow uses 128 channels instead of 512 of the base CNN-VEC and CNN-GAP networks, and trained with L2 regularization strength $\lambda = 10^{-7}$. *Crop* transformation uses zero-padding while *Translate* transformation uses circular boundary condition after shifting images. Each transformation is applied to the test set inputs where shift direction is chosen randomly. Each points correspond to average accuracy over 20 random seeds. FCN had 2048 hidden units.

**Figure 11, Table 1**: For all data augmentation ensembles, first instance is taken to be from non-augmented training set. Further details on kernel ensemble is described in §E. For all kernels, inputs are preprocessed with optimal ZCA regularization observed in Figure 9 (10 for FCN, 1 for CNN-VEC, CNN-GAP and Myrtle.). We ensemble over 50 different augmented draws for FCN and CNN-VEC, whereas for CNN-GAP, we ensemble over 32 draws of augmented training set.

**Figure S3, Table S2**: Details for MSE trials are same as Figure 1 and Table S1. Trials with softmax-cross-entropy loss was tuned with same hyperparameter range as MSE except that learning rate factor range was $c \in \{1, 4, 8\}$.

**Figure S4**: We present result with NTK parameterized networks with $\lambda = 0$. FCN network is width 1024 with $\eta = 10.0$ for MSE loss and $\eta = 2.0$ for softmax-cross-entropy loss. CNN-GAP uses 256 channels with $\eta = 5.0$ for MSE loss and $\eta = 0.2$ for softmax-cross-entropy loss. Random seed was fixed to be the same across all runs for comparison.

**Figure S9**: NTK pamareterization with $c = 4$ was used for both L2 to zero or initialization. Random seed was fixed to be the same across all runs for comparison.

## D  Noise model

In this section, we provide details on noise model discussed in §3.8. Consider a random $m \times m$ Hermitian matrix $N$ with entries order of $\sigma_n$ which is considered as noise perturbation to the kernel matrix

$$\tilde{K} = K + N. \tag{S3}$$

Eigenvalues of this random matrix $N$ follow Wigner's semi-circle law, and the smallest eigenvalue is given by $\lambda_{\min}(N) \approx -\sqrt{2m}\sigma_n$. When the smallest eigenvalue of $K$ is smaller (in order) than $|\lambda_{\min}(N)|$, one needs to add diagonal regularizer larger than the order of $|\lambda_{\min}(N)|$ to ensure positive definiteness. For estimates, let us use machine precision[5] $\epsilon_{32} \approx 10^{-7}$ and $\epsilon_{64} \approx 2 \times 10^{-16}$ which we use as proxy values for $\sigma_n$. Note that noise scale is relative to elements in $K$ which is assume to be $O(1)$. Naively scaling $K$ by multiplicative constant will also scale $\sigma_n$.

Empirically one can model tail $i^{\text{th}}$ eigenvalues of infinite width kernel matrix of size $m \times m$ as

$$\lambda_i \approx C \frac{m}{i^{\alpha}}. \tag{S4}$$

Note that we are considering $O(1)$ entries for $K$ and typical eigenvalues scale linearly with dataset size $m$. For a given dataset size, the power law observed is $\alpha$ and $C$ is dataset-size independent constant. Thus the smallest eigenvalue is order $\lambda_{\min}(K) \sim Cm^{1-\alpha}$.

In the noise model, we can apply Weyl's inequality which says

$$\lambda_{\min}(K) - \sqrt{2m}\sigma_n \leq \lambda_{\min}(\tilde{K}) \leq \lambda_{\min}(K) + \sqrt{2m}\sigma_n. \tag{S5}$$

Consider the worst-case where negative eigenvalue noise affecting the kernel's smallest eigenvalue. In that case perturbed matrices minimum eigenvalue could become negative, breaking positive semi-definiteness(PSD) of the kernel.

This model allows to predict critical dataset size ($m^*$) over which PSD can be broken under specified noise scale and kernel eigenvalue decay. With condition that perturbed smallest eigenvalue becomes negative

$$Cm^{1-\alpha} \lesssim \sqrt{2m}\sigma_n, \tag{S6}$$

we obtain

$$m^* \gtrsim \begin{cases} \left(\frac{C}{\sqrt{2}\sigma_n}\right)^{\frac{2}{2\alpha-1}} & \text{if } \alpha > \frac{1}{2} \\ \infty & \text{else} \end{cases} \tag{S7}$$

When PSD is broken, one way to preserve PSD is to add diagonal regularizer (§3.7). For CIFAR-10 with $m = 50k$, typical negative eigenvalue from `float32` noise is around $4 \times 10^{-5}$ and $7 \times 10^{-14}$ with `float64` noise scale, considering $\sqrt{2m}\sigma_n$. Note that Arora et al. [26] regularized kernel with regularization strength $5 \times 10^{-5}$ which is on par with typical negative eigenvalue introduced due to `float32` noise. Of course, this only applies if kernel eigenvalue decay is sufficiently fast that full dataset size is above $m^*$.

We observe that `FCN` and `CNN-VEC` kernels with small $\alpha$ would not suffer from increasing dataset-size under `float32` precision. On the other-hand, worse conditioning of `CNN-GAP` not only affects the training time (§3.9) but also required precision. One could add sufficiently large diagonal regularization to mitigate effect from the noise at the expense of losing information and generalization strength included in eigen-directions with small eigenvalues.

## E   Data augmentation via kernel ensembling

We start considering general ensemble averaging of predictors. Consider a sequence of training sets $\{\mathcal{D}_i\}$ each consisting of $m$ input-output pairs $\{(x_1, y_1), \ldots, (x_m, y_m)\}$ from a data-generating distribution. For a learning algorithm, which we use NNGP/NTK inference for this study, will give prediction $\mu(x^*, \mathcal{D}_i)$ of unseen test point $x^*$. It is possible to obtain better predictor by averaging output of different predictors

$$\hat{\mu}(x^*) = \frac{1}{E} \sum_i^E \mu(x^*, \mathcal{D}_i), \tag{S8}$$

where $E$ denotes the cardinality of $\{\mathcal{D}_i\}$. This ensemble averaging is simple type of committee machine which has long history [136, 137]. While more sophisticated ensembling method exists (e.g. [138–143]), we strive for simplicity and considered naive averaging. One alternative we considered is generalizing average by

$$\hat{\mu}_w(x^*) = \frac{1}{E} \sum_i^E w_i \, \mu(x^*, \mathcal{D}_i), \tag{S9}$$

were $w_i$ in general is set of weights satisfying $\sum_i w_i = 1$. We can utilize posterior variance $\sigma_i^2$ from NNGP or NTK with MSE loss via Inverse-variance weighting (IVW) where weights are given as

$$w_i = \frac{\sigma_i^{-2}}{\sum_j \sigma_j^{-2}}. \tag{S10}$$

Figure S1: **The CNN-GAP architecture has poor kernel conditioning (a)** Eigenvalue spectrum of infinite network kernels on 10k datapoints. Dashed lines are noise eigenvalue scale from `float32` precision. Eigenvalue for CNN-GAP's NNGP decays fast and negative eigenvalue may occur when dataset size is $O(10^4)$ in `float32` but is well-behaved with higher precision. **(b-c)** Critical dataset size as function of eigenvalue decay exponent $\alpha$ or noise strength $\sigma_n$ given by Equation 1.

In simple bagging setting [139], we observe small improvements with IVW over naive averaging. This indicates posterior variance for different draw of $\{\mathcal{D}_i\}$ was quite similar.

Application to data augmentation (DA) is simple as we consider process of generating $\{\mathcal{D}_i\}$ from a (stochastic) data augmentation transformation $\mathcal{T}$. We consider action of $\mathcal{T}(x, y) = T(x, y)$ be stochastic (e.g. $T$ is a random crop operator) with probability $p$ augmentation transformation (which itself could be stochastic) and probability $(1 - p)$ of $T = \text{Id}$. Considering $\mathcal{D}_0$ as clean un-augmented training set, we can imagine dataset generating process $\mathcal{D}_i \sim \mathcal{T}(\mathcal{D}_0)$, where we overloaded definition of $\mathcal{T}$ on training-set to be data generating distribution.

For experiments in §3.12, we took $T$ to be standard augmentation strategy of horizontal flip and random crop by 4-pixels with augmentation fraction $p = 0.5$ (see Figure S12 for effect of augmentation fraction on kernel ensemble). In this framework, it is trivial to generalize the DA transformation to be quite general (e.g. learned augmentation strategy studied by Cubuk et al. [116, 117]).

## F   ZCA whitening

Consider $m$ (flattened) $d$-dimensional training set inputs $X$ (a $d \times m$ matrix) with data covariance

$$\Sigma_X = \frac{1}{d} X X^T . \tag{S11}$$

The goal of whitening is to find a whitening transformation $W$, a $d \times d$ matrix, such that the features of transformed input

$$Y = WX \tag{S12}$$

are uncorrelated, e.g. $\Sigma_Y \equiv \frac{1}{d} Y Y^T = I$. Note that $\Sigma_X$ is constructed only from training set while $W$ is applied to both training set and test set inputs. Whitening transformation can be efficiently

Figure S2: **Illustration of ZCA whitening.** Whitening is a linear transformation of a dataset that removes correlations between feature dimensions, setting all non-zero eigenvalues of the covariance matrix to 1. ZCA whitening is a specific choice of the linear transformation that rescales the data in the directions given by the eigenvectors of the covariance matrix, but without additional rotations or flips. *(a)* A toy 2d dataset before and after ZCA whitening. Red arrows indicate the eigenvectors of the covariance matrix of the unwhitened data. *(b)* ZCA whitening of CIFAR-10 images preserves spatial and chromatic structure, while equalizing the variance across all feature directions. Figure reproduced with permission from Wadia et al. [112]. See also §3.10.

computed by eigen-decomposition[6]

$$\Sigma_X = UDU^T \tag{S13}$$

where $D$ is diagonal matrix with eigenvalues, and $U$ contains eigenvector of $\Sigma_X$ as its columns.

With this ZCA whitening transformation is obtained by following whitening matrix

$$W_{\text{ZCA}} = U\sqrt{\left(D + \epsilon\frac{tr(D)}{d}I_d\right)^{-1}}U^T. \tag{S14}$$

Here, we introduced trivial reparameterization of conventional regularizer such that regularization strength $\epsilon$ is input scale invariant. It is easy to check $\epsilon \to 0$ corresponds to whitening with $\Sigma_Y = I$. In §3.10, we study the benefit of taking non-zero regularization strength for both kernels and finite networks. We denote transformation with non-zero regularizer, ZCA regularization preprocessing. ZCA transformation preserves spatial and chromatic structure of original image as illustrated in Figure F. Therefore image inputs are reshaped to have the same shape as original image.

In practice, we standardize both training and test set per (RGB channel) features of the training set before and after the ZCA whitening. This ensures transformed inputs are mean zero and variance of order 1.

## G   MSE vs Softmax-cross-entropy loss training of neural networks

Our focus was mainly on fininte networks trained with MSE loss for simple comparison with kernel methods that gives closed form solution. Here we present comparison of MSE vs softmax-cross-entropy trained networks. See Table S2 and Figure S3.

## H   Comment on batch size

Correspondence between NTK and gradient descent training is direct in the full batch gradient descent (GD) setup (see [64] for extensions to mini-batch SGD setting). Therefore base comparison between finite networks and kernels is the full batch setting. While it is possible to train our base models with GD, for full CIFAR-10 large emprical study becomes impractical. In practice, we use mini-batch SGD with batch-size 100 for FCN and 40 for CNNs.

We studied batch size effect of training dynamics in Figure S4 and found that for these batch-size choices does not affecting training dynamics compared to much larger batch size. Shallue et al. [144], McCandlish et al. [145] observed that universally for wide variety of deep learning models there are batch size beyond which one could gain training speed benefit in number of steps. We observe that maximal useful batch-size in workloads we study is quite small.

Figure S3: **MSE trained networks are competitive while there is a clear benefit to using Cross-entropy loss**

Table S2: Effects of MSE vs softmax-cross-entropy loss on base networks with various interventions

| Architecture | Type | Param | Base | +LR+U | +L2+U | +L2+LR+U | Best |
|---|---|---|---|---|---|---|---|
| FCN | MSE | STD | 47.82 | 49.07 | 49.82 | 55.32 | 55.90 |
| | | NTK | 46.16 | 49.17 | 54.27 | 55.44 | 55.44 |
| | XENT | STD | 55.01 | 57.28 | 53.98 | 57.64 | 57.64 |
| | | NTK | 53.39 | 56.59 | 56.31 | 58.99 | 58.99 |
| | MSE+DA | STD | 65.29 | 66.11 | 65.28 | 67.43 | 67.43 |
| | | NTK | 61.87 | 62.12 | 67.58 | 69.35 | 69.35 |
| | XENT+DA | STD | 64.15 | 64.15 | 67.93 | 67.93 | 67.93 |
| | | NTK | 62.88 | 62.88 | 67.90 | 67.90 | 67.90 |
| CNN-VEC | MSE | STD | 56.68 | 63.51 | 67.07 | 68.99 | 68.99 |
| | | NTK | 60.73 | 61.58 | 75.85 | 77.47 | 77.47 |
| | XENT | STD | 64.31 | 65.30 | 64.57 | 66.95 | 66.95 |
| | | NTK | 67.13 | 73.23 | 72.93 | 74.05 | 74.05 |
| | MSE+DA | STD | 76.73 | 81.84 | 76.66 | 83.01 | 83.01 |
| | | NTK | 83.92 | 84.76 | 84.87 | 85.63 | 85.63 |
| | XENT+DA | STD | 81.84 | 83.86 | 81.78 | 84.37 | 84.37 |
| | | NTK | 86.83 | 88.59 | 87.49 | 88.83 | 88.83 |
| CNN-GAP | MSE | STD | 80.26 | 80.93 | 81.10 | 83.01 | 84.22 |
| | | NTK | 80.61 | 82.44 | 81.17 | 82.43 | 83.92 |
| | XENT | STD | 83.66 | 83.80 | 84.59 | 83.87 | 83.87 |
| | | NTK | 83.87 | 84.40 | 84.51 | 84.51 | 84.51 |
| | MSE+DA | STD | 84.36 | 83.88 | 84.89 | 86.45 | 86.45 |
| | | NTK | 84.07 | 85.54 | 85.39 | 86.68 | 86.68 |
| | XENT+DA | STD | 86.04 | 86.01 | 86.42 | 87.26 | 87.26 |
| | | NTK | 86.87 | 87.31 | 86.39 | 88.26 | 88.26 |

# I   Addtional tables and plots

**(a)**

**(b)**

**(c)**

**(d)**

Figure S4: **Batch size does not affect training dynamics for moderately large batch size.**

Table S3: **CIFAR-10 classification mean squared error(MSE) for nonlinear and linearized finite neural networks, as well as for NTK and NNGP kernel methods**. Starting from `Base` network of given architecture class described in §2, performance change of **centering** (+C), **large learning rate** (+LR), allowing **underfitting** by early stopping (+U), input preprocessing with **ZCA regularization** (+ZCA), multiple initialization **ensembling** (+Ens), and some combinations are shown, for **Standard** and **NTK** parameterization. See also Table S1 and Figure 1 for accuracy comparison.

| | Param | Base | +C | +LR | +L2 | +L2 +U | +L2 +LR | +L2 +LR +U | +ZCA | Best w/o DA | +Ens | +Ens +C | +DA +U | +DA +L2 +LR +U |
|---|---|---|---|---|---|---|---|---|---|---|---|---|---|---|
| FCN | STD | 0.0443 | 0.0363 | 0.0406 | 0.0411 | 0.0355 | 0.0337 | 0.0329 | 0.0483 | 0.0319 | 0.0301 | 0.0304 | 0.0267 | 0.0242 |
| | NTK | 0.0465 | 0.0371 | 0.0423 | 0.0338 | 0.0336 | 0.0308 | 0.0308 | 0.0484 | 0.0308 | 0.0300 | 0.0302 | 0.0281 | 0.0225 |
| CNN-VEC | STD | 0.0381 | 0.0330 | 0.0340 | 0.0377 | 0.0279 | 0.0340 | 0.0265 | 0.0383 | 0.0265 | 0.0278 | 0.0287 | 0.0228 | 0.0183 |
| | NTK | 0.0355 | 0.0353 | 0.0355 | 0.0355 | 0.0231 | 0.0246 | 0.0227 | 0.0361 | 0.0227 | 0.0254 | 0.0278 | 0.0164 | 0.0143 |
| CNN-GAP | STD | 0.0209 | 0.0201 | 0.0207 | 0.0201 | 0.0201 | 0.0179 | 0.0177 | 0.0190 | 0.0159 | 0.0172 | 0.0165 | 0.0185 | 0.0149 |
| | NTK | 0.0209 | 0.0201 | 0.0195 | 0.0205 | 0.0181 | 0.0175 | 0.0170 | 0.0194 | 0.0161 | 0.0163 | 0.0157 | 0.0186 | 0.0145 |

| | Param | Lin Base | +C | +L2 | +L2 +U | +Ens | +Ens +C | NTK | +ZCA | +DA +ZCA | NNGP | +ZCA | +DA +ZCA |
|---|---|---|---|---|---|---|---|---|---|---|---|---|---|
| FCN | STD | 0.0524 | 0.0371 | 0.0508 | 0.0350 | 0.0309 | 0.0305 | 0.0306 | 0.0302 | - | 0.0309 | 0.0308 | 0.0297 |
| | NTK | 0.0399 | 0.0366 | 0.0370 | 0.0368 | 0.0305 | 0.0304 | 0.0305 | 0.0302 | 0.0298 | | | |
| CNN-VEC | STD | 0.0436 | 0.0322 | 0.0351 | 0.0351 | 0.0293 | 0.0291 | 0.0287 | 0.0277 | - | 0.0286 | 0.0281 | 0.0256 |
| | NTK | 0.0362 | 0.0337 | 0.0342 | 0.0339 | 0.0286 | 0.0286 | 0.0283 | 0.0274 | 0.0273 | | | |
| CNN-GAP | STD | < 0.0272* (Train accuracy 86.22 after 14M steps) | | | | | | 0.0233 | 0.0200 | - | 0.0231 | 0.0204 | 0.0191 |
| | NTK | < 0.0276* (Train accuracy 79.90 after 14M steps) | | | | | | 0.0232 | 0.0200 | 0.0195 | | | |

Figure S5: **On UCI dataset NNGP often outperforms NTK on RMSE.** We evaluate predictive performance of FC NNGP and NTK on UCI regression dataset in the standard 20-fold splits first utilized in [146, 147]. We plot average RMSE across the splits. Different scatter points are varying hyperparameter settings of (depth, weight variance, bias variance). In the tabular data setting, dominance of NNGP is not as prominent across varying dataset as in image classification domain.

Figure S6: **Centering can accelerate training**. Validation (top) and training (bottom) accuracy throughout training for several finite width architectures. See also §3.3 and Figure 3.

Figure S7: **Ensembling base networks causes them to match kernel performance, or exceed it for nonlinear CNNs.** See also §3.3 and Figure 4.

Figure S8: **Performance of nonlinear and linearized networks as a function of L2 regularization for a variety of widths.** Dashed lines are NTK parameterized networks while solid lines are networks with standard parameterization. We omit linearized `CNN-GAP` plots as they did not converge even with extensive compute budget. L2 regularization is more helpful in networks with an NTK parameterization than a standard parameterization

Figure S9: **L2 regularization to initial weights does not provide performance benefit.** (a) Comparing training curves of L2 regularization to either 0 or initial weights. (b) Peak performance of after L2 regularization to either 0 or initial weights. Increasing L2 regularization to initial weights do not provide performance benefits, instead performance remains flat until model's capacity deteriorates.

Figure S10: **Finite width networks generally perform better with increasing width, but** `CNN-VEC` **shows surprising non-monotonic behavior.** See also §3.6 and Figure 6 **L2**: non-zero weight decay allowed during training **LR**: large learning rate allowed. Dashed lines are allowing underfitting (**U**).

Figure S11: **ZCA regularization helps finite network training. (upper)** Standard parameterization, **(lower)** NTK parameterization. See also §3.10 and Figure 9.

Figure S12: **Data augmentation ensemble for infinite network kernels with varying augmentation fraction.** See also §3.12.

## Footnotes

[5] np.finfo(np.float32).eps, np.finfo(np.float64).eps

[6]For PSD matricies, it is numerically more reliable to obtain via SVD.