[Reviews · NeurIPS 2020]

Review 1

Summary and Contributions: UPDATE: I read the other reviews and the authors' response, and I my review stands as is. I think the work is a nice empirical contribution, and I am satisfied with the authors' responses to the questions I asked below. This paper presents a large variety of extensive experimental results comparing kernel methods derived from infinite-width neural networks to the training of finite-width neural networks. The experiments cover fully-connected and convolution network architectures, as well as kernels derived from their corresponding large-width limit. Additionally, the kernels include both the Gaussian process kernel related to using large-width networks as a prior for Bayesian inference, and the neural tangent kernel, related to training a large width network with gradient descent. The most striking results involve kernels derived from fully-connected networks outperforming their corresponding finite-width counterparts, although for convolution architectures the reverse tended to be true.

Strengths: This paper is entirely an empirical paper. As such, while I cannot evaluate or replicate the actual experiments, I can comment on the experimental methodology. The experiments seemed very well designed around testings specific hypotheses as well as attempting to broadly evaluate the merits of finite-width neural networks vs. infinite-dimensional kernels. Furthermore, as the subject of wide network limits is a popular one in the theory of deep learning community, having a wealth of careful experimental results to reference is significant, and therefore I think this paper has the potential to be pretty influential.

Weaknesses: Given the broad list of experiments and careful effort into testing many hypotheses, it's somewhat difficult to find fair weaknesses here. Of course, one can always point to a particular configuration or type of experiment that's not included, but that seems besides the point, especially given the extensive coverage of what _was_ included. Having said that, my main question and criticism has to do with layer dependence. The FCNs studied were 3-layers deep, and the CNNs were 8-layers. I would have liked to have seen some kind of study that compared the performance of the infinite-width kernels and finite-width networks across a variety of layers. I also wonder whether the fact that FCNs performed poorly compared to their kernel counterparts would persist for deeper networks. I bring this up since much of what makes deep learning special is that it's ``deep'' and so I wonder if 3-layers is really sufficient in that regard.

Correctness: Given the empirical nature of the paper, it's hard to directly evaluate its correctness. In terms of the empirical methodology, for the most part I think it was superb (as I discussed in the ``strengths'' section above). One thing that I'm curious about relating to the paper's results is the well-known fact that infinite width networks cannot learn any representations (i.e. the kernels don't depend on the data). On the other hand, a common hypothesis about why neural networks are so powerful is that they are really great at learning useful representations. Given that, it seems like there's some tension with the results of FCNs at finite width underperforming their kernel counterparts. I wonder if the problems studied were too simple to require the learning of useful representations (or the FCNs were too shallow to learn them, given they only had 3 layers). For sufficiently complicated problems and large enough models (especially on problems where the representations are usefully fine-tuned across different tasks) I suspect the finite width networks have to outperform kernels. Perhaps this also explains why for the deeper CNNs the neural networks beat the kernels?

Clarity: The paper is well organized and well written. A large variety of experimental results are presented in a clear and logical manner. Additionally, the plots make it easy to glance at the results and understand the experimental outcomes. Given the number and breadth of the experiments, it shows that care went into the organization and presentation, and that is definitely appreciated.

Relation to Prior Work: Since this work represents an extensive set of experimental results, it's pretty clear how this work differs from previous contributions. Furthermore, the introduction cites a large number of papers on kernel methods, on understanding neural networks, and on the large width limit. In addition to citing important theoretical work on which their experiments are designed to investigate further, the authors are clear to highlight other empirical results, making it clear whether they confirm or extend them.

Reproducibility: Yes

Additional Feedback: UDPATE: I am satisfied with the authors' responses to the questions I asked below. Some comments and questions: On line 67, the authors say that the kernel matrix has 6 x 10^7 squared entries. How do they arrive at this number? What initializations were used for the biases and weights? (I'm curious both for defining the kernel and for the finite width networks.) This should definitely be included in the main body. In Figure 9, there's mention of a ``narrow model with weight decay''. What precisely does narrow mean in this context? Furthermore, it should say in the main body of the paper the width (for FCN) and number of channels (for CNN) used for the majority of the experiments, since that's an important parameter. For the NNGP experiments in 3.2, did the authors consider also using the finite-width corrections to the NNGP computed by Yaida (https://arxiv.org/abs/1910.00019)? I'd be very curious whether the incorporation of such corrections hurts or improves performance.


Review 2

Summary and Contributions: Post-rebuttal: I read the other reviews and the authors' response. I keep my original rating. The paper conducts a careful and large scale empirical comparison between finite and infinite width neural networks through a series of controlled interventions. By doing so, a lot of interesting observations are made. Based on the experiments, the authors also propose an improved layer-wise scaling for weight decay and improve the performance of NNGP and NTK. Overall, I think the experiments in the paper are carefully designed and well-conducted. I believe this paper will have a great impact on understanding and utiltizing infinite width neural networks.

Strengths: 1. A lot of interesting observations are made. For example, the authors show that large learning rates and l2 regularization can break the correspondence between kernel methods and finite width neural networks. Those observations are important in the sense that it provides strong empirical ground for future theoretical or empirical study in this direction. 2. Based on the result that L2 regularization performs much better with NTK parameterization, the authors propose a new layer-wise scaling for weight decay, which does improve the performance empirically. 3. The authors observe a surprising result that performance is not monotonic in width. That is really interesting and might inspire some new theories.

Weaknesses: I can hardly find any weakness of the paper. Great paper!

Correctness: The claims and method look correct to me. The empirical methodology seems flawless.

Clarity: Yes, I really enjoy reading the paper as it is so well-written. The authors divide all observation into subsections, which makes it very easy to follow.

Relation to Prior Work: The relation to prior work is clearly discussed in every small section.

Reproducibility: Yes

Additional Feedback:


Review 3

Summary and Contributions: The authors present a large comparative study of Fully connected, CNN+pooling, CNN+FC neural networks with Gaussian-Process based infinitive methods. Furthermore, common training strategies s.a. L2-regularization, preprocessing with whitening, preprocessing by centering, learning rate, early stopping, ensembling, ... are investigated. No significant methods are introduced. They use CIFAR-10 (and as a check CIFAR-100) as datasets to benchmark on.

Strengths: extensive evaluation

Weaknesses: It is unclear to me why only 8 layer networks with pooling/ linear readout in the last layer are considered. Shallow networks are known for performing badly on CIFAR. E.g. a ResNet50 would perform ~90% on CIFAR-10.

Correctness: not my area of expertise.

Clarity: The paper is structured well given that it is a large listing and discussion of ~12 empirical observations. It is also written clearly and seems - to my limited understanding - very thorough. However, the main underlying concepts of NNGP and NTK are not explained in the main paper. The headings of the main figure 1 are not directly explained. Therefore, for a reader who is not familiar with the field of infinitive neural networks is unable to understand the main claims of the paper.

Relation to Prior Work: not my area of expertise.

Reproducibility: Yes

Additional Feedback:


Review 4

Summary and Contributions: The authors do a careful and thorough empirical comparison between the performance of standard finite networks under SGD and infinite networks --- both NNGPs and NTK.

Strengths: The empirical evaluations are thorough, comprehensive and convincing. The observation about ZCA is particularly cool.

Weaknesses: My key issue is that theoretical work was ArXived last year and just published at ICML with a theoretical view on the differences between finite and infinite networks: https://arxiv.org/abs/1910.08013 Why bigger is not always better: on finite and infinite neural networks This paper explicitly talks about how performance changes as we increase width, including that there should be an optimal width in cases of model mismatch. Otherwise, the paper's main flaw is that it reads as a collection of interesting observations, rather than making a single strong point. In principle however, I don't think this should be a barrier to publication.

Correctness: Extensive, sensible empirical evaluations.

Clarity: Yes.

Relation to Prior Work: Extensive reference list: only issue above.

Reproducibility: Yes

Additional Feedback:

[Author Response · NeurIPS 2020]

We thank the reviewers for their time and constructive feedback to improve the paper.

**R1**: *-Depth dependence*: We agree that understanding the role of depth is another interesting analysis. The main
reason we chose these particular depths was because the performance was near optimal, for a given architecture class.
For FCNs on classifying MNIST / CIFAR-10, we found that depth does not benefit performance as much as in CNNs
and optimal performance is often achieved in shallow networks. This is consistent with [4, Table 1]. On the other
hand, CNNs certainly benefits from depth as shallow networks' receptive fields cannot cover the whole image (with
3x3 filters with stride 1). Note that prior works considering similar architectures [26, Table 1; 40, Tables 1-4] have
observed roughly constant performance of finite and infinite CNNs on CIFAR-10 across depths from 5 to 20, indicating
that increasing depth without other changes in this specific setting would not yield dramatic improvements.

*-Usefulness of learned representation*: While it is widely believed that learned representation are important for deep
learning, and there's implicit evidence based on transfer learning, it is not yet proven to be useful for all deep learning
models. We argue that studying the relationship between infinite and finite neural networks provides a lens to study the
utility of learned representations. Our empirical findings suggest that FCNs tend not to learn a useful representation
by simple SGD training. Moreover, we believe that depth is not a major factor here, at least within the scope of our
experiments, since for FCN both tuned finite and infinite networks show better performance at shallow depth.

*-Matrix size*: Since we compute all pixel-pixel covariance for each pair of inputs for CNN-GAP, the internal matrix
size is $(6 \times 10^4$ (# of train+test images) $\times 32 \times 32$ (# of pixels) $\approx 6 \times 10^7)^2$.

*-Finite width correction of [Yaida 2019]*: This is a very interesting suggestion and could be a future work. As far as
we know, there is not yet an efficient, scalable implementation for the non-Gaussian corrections of [Yaida 2019] which
could be applied to the full CIFAR-10 dataset.

*-Network hyperparameters*: We used (ReLU) critical initialization for weight variance and small bias variance, and we
used 512 channels per layer for CNN base models (128 channels for the "narrow" one, as specified in SM C.1, Figure
9). We will mention these key architectural details described in the SM in the main text.

**R2**: We thank the reviewer for encouraging and positive feedback on our submission!

**R3**: *- Architecture choice*: We agree with the reviewer that one could add various architectural components such as
batch normalization or residual connections to improve finite network performance. As a side note, the best reference
for ResNet on CIFAR10 w/o data augmentation (still with other training tricks) achieves 86.37% [Huang & Sun et
al., Deep Networks with Stochastic Depth, ECCV 2016] which is in a similar ball-park to CNN-GAP architecture we
study w/o data augmentation.) We emphasize that, in this work we strive to find simple architectures where the infinite
width limit is well developed and scalable to carefully study how it relates to corresponding finite networks.

*-No description on NNGP/NTK*: Our paper indeed assumes familiarity with infinite neural networks and made a
judicious choice to not include a review to allow more space for empirical results. We will modify the text to include
specific resource suggestions for readers who want to learn more.

**R4**: *On [Aitchison 2020]*: We thank the reviewer for bringing this reference to our attention. It is definitely relevant
and we agree that it should be included in our related work discussion. However, we want to highlight that the focus
of that paper is distinct and with a more limited scope. [Aitchison 2020] focused on the comparison between finite and
infinite networks in the *pure Bayesian setting*. By contrast, ours focuses on the comparison of the mean prediction of
infinite networks (NNGP/NTK) to finite networks optimized by modern deep learning techniques. In particular we did
a thorough (reverse) ablation analysis (Fig 1), exploring the effects and implicit biases induced by large learning rate,
early stopping, weight decay, data-augmentation, data preprocessing (ZCA), parameterization methods, and readout
strategies (pooling vs vectorization).

*-Lack of a single strong point*: Due to the nature of our empirical study, we did not intend to make a single strong
point. We want to emphasize that our goal is to perform a *thorough* scientific investigation of finite and infinite
networks, answering questions about the factors of variation that drive performance of neural networks, clarifying
misconceptions in the existing literature, and uncovering a variety of new and surprising phenomena (e.g. nonlinear
interactions between L2-regularization, large learning rate and early stopping). We believe the careful experiments in
our paper provide a service to the deep learning community, and may inspire future theoretical and empirical work.

[Meta-Review · NeurIPS 2020]

This paper conducted thorough experiments to compare performances of finite with and infinite width networks. The network architectures investigated are FNN, CNN with/without global average pooling (GAP), and two parameterizations of them (standard one and NTK one) were compared. Several techniques such as regularization and ensemble learning are applied to these methods. Throughout the experiments under several different settings, they derived several conclusions from several view points. They also developed best practices for using non-trainable kernels on the CIFAR-10 classification task. The authors conducted thorough experiments in depth. The obtained insights will be quite useful for practitioners and promote further investigations of infinite width networks. Thus, this study is beneficial for the community.